# Public Opinion on Food Policies to Combat Obesity in Spain

**DOI:** 10.3390/ijerph19148561

**Published:** 2022-07-13

**Authors:** Cristina Cavero Esponera, Sara Fernández Sánchez-Escalonilla, Miguel Ángel Royo-Bordonada

**Affiliations:** 1Department of Preventive Medicine and Public Health, San José and Santa Adela Red Cross Hospital, 28003 Madrid, Spain; 2Department of Preventive Medicine and Public Health, Albacete University Teaching Hospital Complex, 02006 Albacete, Spain; sarafse7@gmail.com; 3National School of Public Health, Institute of Health Carlos III, 28029 Madrid, Spain; mroyo@isciii.es

**Keywords:** public opinion, overweight, obesity, policy, regulation, Spain

## Abstract

(1) Introduction: Poor diet is the fourth-leading cause of mortality in Spain, where adherence to the Mediterranean diet has declined in recent decades. To remedy this situation, a series of food policies have been proposed that would be easier to implement if they had public support. (2) Material and methods: Cross-sectional study covering a representative sample of the Spanish population (*n* = 1002), using telephone interviews that addressed nine food policies under four headings, namely, advertising, labeling, composition, and provision and sale. The sociodemographic determinants of support for these policies were analyzed using the chi-squared (χ^2^) test and Poisson multiple regression models with robust variance. (3) Results: All the proposed measures received more than 60% support. The policies that received greatest support were those targeting labeling at 96.6%, while the policies that received the least support were those directed at banning free refills at restaurants, at 63%. Support for policies was higher among women, older adults, and persons professing left-wing ideological affiliations. Compared with men, women’s support for advertising policies was 21% higher: similarly, compared with the youngest age group (18–29 years), support by the over-65 segment for provision and sale policies was 52% higher. Support for composition policies was 28% lower among persons with right-wing as opposed to left-wing political sympathies. (4) Conclusions: The authorities enjoy the support of the Spanish public as regards implementing food policies proposed by experts and overcoming the resistance of sectors opposed to such measures.

## 1. Introduction

Poor diet is the third-leading cause of mortality in the world and the fourth-leading cause in Spain [1]. Unhealthy eating patterns are characterized by a high intake of unhealthy foods and beverages (UF&B: products high in calories, sugar, salt, and low-quality fats; poor in fiber and essential micronutrients; and usually ultraprocessed) to the detriment of healthy foods such as fruit and vegetables, legumes, nuts, grains, fish, and yogurt. Furthermore, the consumption of UF&B contributes to overweight and abdominal obesity, which affect 58% and 65% of the Spanish population, respectively [2], and constitute the fifth-leading cause of mortality.

In recent decades, Spain has seen a decline in adherence to the Mediterranean diet [3]. The percentage of calories from ultraprocessed foods and beverages in Spanish home grocery purchases tripled across the period 1990–2010, rising to over 30% [4]. In the last decade, the consumption of candies and sugary desserts has risen, whereas that of fruit, vegetables, and fish has fallen [5]. These changes have been more intense among the unemployed and persons with a low education level, thereby increasing the social inequalities in healthy eating habits. Similarly, children and adolescents from families with lower purchasing power show a lower consumption of fruit and vegetables and a higher intake of candies, fast foods, and sugar-sweetened beverages [6].

The mass production and distribution of UF&B, coupled with low prices and sophisticated marketing campaigns to promote their consumption, are the main factors driving this progressive deterioration in eating habits [7]. To confront this problem, experts advocate a unified approach to its determinants involving implementing a raft of food-and-drink pricing, advertising, provision, composition, and labeling policies [8]. The World Health Organization (WHO) European Region has urged Member States to introduce these policies to combat the epidemic of obesity and noncommunicable diseases associated with unhealthy diet [9]. Although there have been some advances in fiscal policies, school food catering, and the reformulation of products, there is nevertheless a need for more ambitious approaches that would also include the regulation of food advertising and front-of-pack nutrition labeling [10]. In Spain, with a nutrition strategy based on educational measures and public/private collaborations embodied in self-regulation agreements [11], these advances have been very timid, and to date, pressure from the agri-food and advertising industries has succeeded in blocking the more ambitious proposals, such as the levying of a tax on sugar-sweetened beverages, the regulation of food advertising, or the implementation of a front-of-pack nutrition labeling system [12].

Public opinion is a factor that affects the viability of policies since politicians show themselves to be more willing to implement a policy if it receives popular support [13]. Consumers of a more liberal persuasion are more in favor of buying at farmers markets and consuming local and seasonal products than are individuals who embrace a more traditional ideology [14]. Even the best of policies may prove difficult to enact or implement in the absence of public support [15]. When the news media reflect certain industries’ influence on and responsibility for population eating habits, the public tends to support the government regulation of these sectors, e.g., Spain’s support for tobacco control policies grew as public opinion shifted from a predominant view of tobacco use as a free choice to the notion of tobacco as an addictive product aggressively marketed and manipulated by the tobacco industry [16]. Support for food policies not only increases when the population is aware of the harmful effects of unhealthy diets and the environmental causes of obesity [17] but also depends on sociodemographic factors, with older adults, women, and persons with university or higher education being more favorable toward the implementation of such policies [18,19]. Population support for food policies is also conditioned by what the policy deals with: labeling policies and restrictions on unhealthy food advertising to children, more related to informing and educating the population or protecting vulnerable groups, receive stronger support than policies involving price raising [15,17].

To our knowledge, this is the first study to analyze the levels of public support in Spain for a raft of food policies. In a previous paper, we analyzed the degree to which Spanish society supported the imposition of taxes on sugar-sweetened beverages as a means of combating overweight and obesity [20]. This paper now reports an analysis of public opinion on food-and-drink pricing, advertising, provision, composition, labeling, and sale policies.

## 2. Material and Methods

### 2.1. Survey Sample

We conducted a cross-sectional study, interviewing persons aged 18 years and over who were resident in Spain. As its base, the initial sampling framework targeted homes in Spain with a landline telephone installed in September 2018. Of the total of 99.6% of homes that had a telephone, 23.9% only had a cell phone link, 1.6% only had a landline link, and 74.2% had both [21]. To extend the study’s coverage to persons who did not possess a landline telephone or whose names were not shown in the database at their own request, a cell phone database was incorporated into the sampling framework, thus establishing a 50–50 distribution between landline and cell phone numbers. The cell phone database was created with randomly generated numbers starting with 6 and 7, deleting the prefixes (the first 3 digits of cell phone numbers) that do not exist. 

The sample was obtained using random sampling stratified by size of habitat and Autonomous Region (Comunidad Autónoma). The selection of homes as first-stage sampling units allowed us to assume simple random sampling in each geographical stratum by using the database of households with a telephone (or mobile number). The proposed stratification by sex and age meant that these two variables would also have to be taken into account in the selection process of the final stage sampling units, the participating individuals. The challenge was that the stratum to which a sampling unit belonged was not known until after data collection. The sizes of the strata were obtained a priori from official statistics, but the sample units could not be classified into strata until after the sample data were known. Post-stratification by sex and age consisted of selecting the sample elements by means of simple random sampling in the household and classifying them a posteriori until reaching the predetermined sample size in each stratum, according to data from official statistics. This task was automatically performed with the aid of Bellview CATI 6, Confirmit, Oslo (Norway) (computer-assisted telephone interviewing) software. The survey was designed to obtain a 95.5% confidence level, with a precision of ±3.5% for an estimated proportion of 50%. The response rate was 76%, thus making it necessary to select a total of 1319 individuals until the preestablished sample size was reached. The final sample totaled 1002 participants with proportional allocation per stratum.

### 2.2. Survey Questionnaire Administration

The study questionnaire was purpose-designed by the study researchers referring to other questionnaires used in similar interviews as reference [15,17,19,22,23] and then sent to public health policy experts and representatives of the Food and Agriculture Organization (FAO) and the WHO, whose suggestions were subsequently incorporated. To ascertain the appropriateness, comprehensibility, and order of the questions; the length and duration of the questionnaire; and the level of response, we carried out a pilot study on a sample of 60 persons from 30 May through 6 June 2018. Due to difficulties of comprehension or inconsistencies in responses, the wording of three questions was amended halfway through the field work of this pilot study. 

The questionnaire comprised 40 questions structured in several sections. The section designed to assess the level of support for food policies using a 5-point Likert-type scale (ranging from “strongly agree” to “strongly disagree”) was made up of 12 questions grouped under the following headings: provision and sale (two measures); labeling (one measure); composition (two measures); and advertising of UF&B (four measures). For each proposed measure, a variable with two categories was constructed: “agree” if the participants agreed or strongly agreed with it and “disagree” in the remaining cases. Likewise, for each food policy heading, a variable with two categories was constructed: “agree” if the participants agreed or strongly agreed with all the measures envisaged under a specific heading and “disagree” in the remaining cases. The section on perceived health included questions on anthropometric data (weight and height), physical activity, sleep, and food. Lastly, the section on sociodemographic information recorded sex, age, nationality, education level, marital status, occupational status, political orientation, and occupation, which served to assign social class [24]. The interviews were held across the period 10 September through 1 October 2018, using a CATI with a mean duration of 20 min, administered by trained interviewers.

### 2.3. Statistical Analysis

We performed a descriptive analysis using the frequency distribution of the sociodemographic variables. Support for food policies was ascertained by calculating the percentage who agreed or strongly agreed with each specific measure on the one hand and with all the measures under each heading (provision and sale, labeling, composition, and advertising) on the other. To compare the levels of support for the proposed measures according to the sociodemographic variables, the chi-squared (χ^2^) test was applied. To analyze the determinants of the levels of support for the food policies under each of the four headings, we used Poisson regression models adjusted for the sociodemographic variables. To correct small deviations in the final valid sample with respect to the proportional allocation, a weighting coefficient for each case was applied in all the calculations, taking into account the proportional distribution by the variables of sex, age, Autonomous Region, and habitat.

## 3. Results

### 3.1. Sociodemographic Characteristics of the Study Sample, Representative of the Spanish Adult Population, 2018

Table 1 shows the sociodemographic characteristics of the sample. The mean age of the 1002 survey participants was 50.3 years, and 47.3% were men. The breakdown showed the following: persons aged 18 to 29 years accounted for 15.3% of the sample versus those aged over 65, who accounted for 23.9%; 42.4% had low socioeconomic status; and over half had secondary or higher education (57.5%), were gainfully employed (53.2%), and were ideologically aligned with the political center (50.4%).

### 3.2. Percentage Agreement with Food Policies

Table 2 shows the degree of agreement with food policies designed to promote a healthy diet. All the proposed measures received majority (over 60%) support from the public. Labeling policies were the measure that received the highest degree of support, with 96.6% of participants agreeing or strongly agreeing, followed by the policies related to limiting salt, sugar, and unhealthy fat content, with 92.2% support; 89% of participants reported being in favor of banning UF&B from being allowed to make nutritional or health claims, and 85.4% supported a ban on advertisements of these products targeted at children. The ban on advertising UF&B at sport events and a ban on refills were among the least popular measures, with 70.2% and 63% support, respectively. All measures received more support among women and older adults (*p* < 0.05) except on the one hand, the banning of UF&B advertising targeted at children; the limitation on the portion sizes of UF&B; and banning UF&B in schools, sports events, or health facilities, which received similar degrees of support in both sexes; and on the other, the policies aimed at labeling, banning UF&B from being allowed to claim nutritional or health benefits, and limiting the content of essential nutrients, where no differences were observed by age. Participants with a university education were more inclined to support a ban on UF&B being allowed to carry nutritional or health claims (92.1%; *p* < 0.01). Students were the participants least in favor of the proposed measures, with under 50% supporting a ban on handouts of toys and gifts on children’s menus, free refills of sugar-sweetened beverages at restaurants, and UF&B advertising at sports events (*p* < 0.01). Lastly, participants with left-wing ideological views showed greater support for a ban on UF&B advertising targeted at children and handouts of toys and gifts on children’s menus (90.4% and 81.4%, respectively; *p* < 0.01).

### 3.3. Prevalence Ratios (95% CI) of Support for Food Policies, Yielded by Poisson Regression Models

Table 3 shows the prevalence ratios (PRs) of support for all measures under each food policy heading, yielded by adjusted Poisson regression models. Support for all policies was higher among women than men, ranging from 7% higher for labeling policies (PR = 1.07; CI = 1.01–1.13) to 21% higher for advertising policies (PR = 1.21; CI = 1.06–1.37), though the difference was not statistically significant for provision and sale policies (*p* = 0.09). Compared with the youngest age group (18–29 years), support for advertising and provision and sale policies was 27% (*p* = 0.02) and 52% (*p* = 0.01) higher, respectively, among the over-65 segment. Support for composition policies was 28% lower in persons professing right-wing as opposed to left-wing ideological affiliations (PR = 0.72; CI = 0.56–0.94), while support for the provision and sale policies was 26% higher among participants of middle socioeconomic status rather than higher status (PR = 0.74; CI = 0.60–0.90).

## 4. Discussion

The majority of the Spanish population support food-and-drink advertising, composition, labeling, and provision and sale policies. Around 9 out of 10 Spanish citizens agree on banning UF&B advertising targeted at children; limiting the salt, sugar, and unhealthy fat content of foods and drinks; and including a traffic light or health warnings on the labels of products that contain these ingredients in excess. Around 8 out of 10 Spanish citizens are committed to limiting the size of sugar-sweetened beverages and support bans on both on the sale of UF&B at primary schools, sport events, or health facilities and on handouts of toys and gifts on children’s menus; 6 to 7 of every 10 citizens are committed to banning free refills at restaurants. While support for food policies was higher among women, older adults, and persons with left-wing ideological affiliations, support for most of the proposed measures was noticeably lower among students and persons with right-wing ideological affiliations.

This is the first study in Spain to analyze the degree of support for a raft of food policies among a population-representative sample. Policies targeted at the child population, as well as less intrusive policies such as those relating to nutritional labeling and health warnings, which are limited to informing consumers about the nutritional properties of a product and its effects on health, were the ones that received the highest level of public support, in line with what is reported in the literature [19]. A total of 85.4% of the Spanish population was in favor of banning UF&B advertising targeted at children, the highest figure for any country in the region, where support ranges from approximately 50% in Denmark to 60% in Italy, Belgium, and the United Kingdom and 80% in Germany, a figure similar to that reported in Australia [15]. Support for a nutritional traffic light and health warnings on food-and-drink labeling exceeded 90%, once again the highest figure for any country in Europe, followed closely by Italy and Germany at 87%. On establishing comparisons, it should be borne in mind that support for food policies increases with the public’s knowledge about the harmful effects of UF&B on their health [25,26]. In the years preceding this survey, the Spanish Healthy Food Alliance (Alianza Española por la Alimentación Saludable) launched a number of dissemination campaigns about the effects of ultraprocessed foods and drinks on health, which culminated in the so-called Defiéndeme campaign in 2018 to ask for a ban on the advertising of all foods harmful to children’s health [27]. Likewise, the nutrition group of the Spanish Society for Epidemiology (Sociedad Española de Epidemiología) drew up a number of policy briefs that culminated in the publication of a paper that demanded the implementation of the principal food policies covered in our survey [28]. These movements have served to boost public support for the proposed measures.

In our study, support for the proposed measures under the four food-policy headings was higher among women, a finding in line with what has repeatedly been observed in the literature [15,29,30]. This could be due to the fact that in recent years, there has been growing concern among consumers about the nutritional quality of the products they consume, with this phenomenon being more marked among women, who are more deeply involved in following good dietary habits and maintaining healthy lifestyles [18,19,31]. Indeed, the European Health Interview Survey shows that Spanish women register a higher intake of fruit and vegetables and do more low-to-moderate intensity physical activity than men [32]. Moreover, women are more exposed to the social consequences of food-related diseases, such as obesity, which is a cause of discrimination in the workplace among European women in general and Spanish women in particular [33]. Diseases related to unhealthy eating habits tend to present in adult ages and are more prevalent among older adults [2,4], thus leading to greater motivation to follow a healthy diet and, in turn, perhaps accounting for the higher support for food policies observed in older age groups, as shown in other studies [18,19].

The higher the education level achieved, the better the knowledge of the harmful effects of consuming UF&B, something that could account for the greater support for food policies observed among university graduates in the various European countries [30,34,35]. In our study, participants who had a university education were more likely to support the limitation of the content of unhealthy nutrients, UF&B advertising targeted at children, and the use of nutritional and health claims in UF&B, though the differences were only statistically significant in this last case. Hence, promoting education campaigns about the harm caused by UF&B might increase public support for food policies [36]. In our study, less support was observed for provision and sale policies among groups of middle and lower socioeconomic status, in line with other studies conducted elsewhere [18,22,37,38] and in Spain [25] that report lower support for food policies in this segment, especially where taxes are involved. A systematic review shows scant concordance in support for policies related with diet and physical activity according to socioeconomic status: Of a total of seven studies analyzed, there were two in which groups of a low socioeconomic level felt favorable toward the interventions and three in which this group felt less favorable, and in the remainder, no differences were observed [19].

The lower support for measures targeted at regulating the food environment among participants reporting center-leaning and right-wing political affiliations is in line with the results of many studies [15,22]. This might be related to the dominant narrative [39], backed by the mass communication media [39,40], that eating habits are above all a matter of individual choice and responsibility [41]. Even so, scientific evidence shows that the main determinants of diet are social and environmental factors. Hence, educational measures targeted at acting on the individual are prioritized, and more interventionist, market-regulation measures are ruled out, in line with more liberal and conservative thinking [24,42,43,44].

Understanding the differences in the support for each food policy across sociodemographic characteristics could help with designing communication campaigns aimed at less supportive populations to counter the dominant narrative, focusing on the external causes of obesity (social and commercial factors) and solutions, as suggested by Mazzocchi et al. [15]. In addition, to prompt the public to accept food policies, we suggest developing education programs on nutrition for the less educated and less supportive groups. Furthermore, to foster food policies and overcome resistance, it will be necessary to create alliances between public health professionals and civil society organizations. Enacting food policies will, in turn, change public attitudes as a result of the preference for the current state of affairs, the so-called status quo bias, or by a process of cognitive dissonance whereby attitudes follow behavior [19].

### Limitations and Strengths

This is the first study to analyze the degree of support for a raft of food policies among a representative Spanish population sample. The principal study limitation lies in a possible non-response bias: 24% of the individuals selected refused to participate. As a bias-correction technique, we used semicontrolled sampling replacement, with which corrected prevalence estimates were obtained similar to those used with other methods [44]. Although this technique ensured that the sample would continue to be representative of the Spanish population in terms of the sociodemographic characteristics used for sampling purposes, it is nonetheless possible that the participants most motivated to respond might have differed from those who did not respond in terms of other characteristics that determine support for food policies. A further limitation concerns the sample size, which although it was large enough for calculating the prevalence of support for food policies was nonetheless limited for studying small-magnitude associations with sufficient accuracy. Moreover, the fact that we carried out a cross-sectional study meant that the causality of the associations observed could not be established. Another possible bias is that of social desirability, but the level of support for the proposed measures was so high that even ignoring this possible bias, the food policies would continue to receive majority support among the Spanish population.

## 5. Conclusions

This study highlights wide-ranging Spanish public support for the implementation of food policies aimed at regulating the advertising, labeling, composition, and provision and sale of UF&B. These results successfully counter the arguments raised against these food policies, which appeal to individual freedoms and preferences, and the health authorities can therefore rely on them to implement the policies proposed by nutrition and public health experts and overcome the resistance of the sectors opposed to such measures. Food policies recommended by the WHO, like food advertising, price policies, front-of-pack nutrition labeling, and UF&B bans in school and health care facilities, received the support of the majority of the Spanish population and should be prioritized.

## Figures and Tables

**Table 1 ijerph-19-08561-t001:** Sociodemographic characteristics of the study sample, representative of the Spanish adult population, 2018.

	*n*	%
Total	1002	100
Sex		
Men	474	47.3
Women	528	52.7
Age (years)		
>65	240	23.9
45–64	358	35.7
30–44	251	25.0
18–29	153	15.3
Education level		
University	271	27.0
Secondary	576	57.5
Primary	155	15.5
Occupational status (*n* = 1001)		
Gainfully employed	533	53.2
Pensioner	255	25.4
Unemployed/unremunerated work	146	14.6
Student	67	6.7
Ideology (*n* = 855)		
Left-wing	319	37.3
Center	431	50.4
Right-wing	105	12.3
Social class (*n* = 922)		
High	273	29.6
Middle	258	28.0
Low	391	42.4

**Table 2 ijerph-19-08561-t002:** Percentage agreement of the Spanish population with food-and-drink advertising and labeling policies, 2018.

	Advertising Policies								Labeling Policies		Composition Policies				Provision and Sale Policies			
	Ban UF&B Advertising Targeted at Children (N = 997)		Ban Handouts of Toys and Gifts with UF&B on Children’s Menus (N = 992)		Ban UF&B Advertising at Sports Events (N = 992)		Ban UF&B from Being Allowed to Claim Nutritional or Health Benefits (N = 999)		Add a TL ** on Food Labeling or Place Health Warnings on UF&B Labels (N = 1002)		Limit Salt, Sugar, and Unhealthy Fat (N = 999)		Limit the Size of UF&B Portions (N = 997)		Ban the Sale of UF&B in Schools and Health Care Facilities (N = 999)		Ban Free Refills of Sugar-Sweetened Beverages at Fast food Restaurants (N = 992)	
	Agree (%)	*p* *	Agree (%)	*p **	Agree (%)	*p **	Agree (%)	*p **	Agree (%)	*p **	Agree (%)	*p **	Agree (%)	*p **	Agree (%)	*p **	Agree (%)	*p **
Total	85.4		75.2		70.2		89.0		96.6		92.9		80.7		82.6		63.0	
Sex		0.53		0.03		0.00		0.01		0.01		0.03		0.35		0.86		0.01
Men	84.7		72.1		63.3		86.3		95.2		91.0		79.2		81.9		58.9	
Women	86.1		78.1		76.8		91.5		98.0		94.6		81.6		82.4		66.8	
Age (years)		0.02		0.00		0.00		0.38		0.47		0.11		0.07		0.00		0.00
18–29	77.7		53.8		58.8		86.5		94.8		95.1		78.8		75.1		45.8	
30–44	84.7		74.9		63.8		90.7		96.4		95.1		76.0		77.7		55.8	
45–64	87.7		82.8		75.8		90.0		96.7		91.9		81.7		84.9		71.2	
>65	88.0		78.3		77.4		86.9		97.9		90.1		85.0		88.1		70.9	
Educational level		0.09		0.93		0.66		0.00		0.56		0.08		0.07		0.46		0.35
Primary	80.6		73.9		73.4		81.5		97.9		88.6		87.3		84.2		67.6	
Secondary	85.2		75.3		69.8		89.5		96.6		93.5		79.6		80.8		63.0	
University	88.6		75.5		69.4		92.1		96.9		93.8		78.3		83.8		60.4	
Occupational status	(*n* = 996)	0.01	(*n* = 991)	0.00	(*n* = 991)	0.00	(*n* = 998)	0.04	(*n* = 1001)	0.01	(*n* = 998)	0.00	(*n* = 996)	0.25	(*n* = 998)	0.00	(*n* = 991)	0.00
Gainfully employed	86.3		77.0		67.9		91.4		96.8		95.3		78.2		80.9		60.7	
Pensioner	86.5		77.6		77.4		85.6		97.6		87.9		83.4		88.0		71.4	
Unemployed/unremunerated work	86.9		77.9		76.2		87.7		97.5		93.2		83.8		82.7		68.5	
Student	70.9		44.7		49.5		83.6		89.7		90.2		79.9		69.6		39.3	
Ideology	(*n* = 852)	0.00	(*n* = 847)	0.00	(*n* = 847)	0.13	(*n* = 854)	0.11	(*n* = 855)	0.66	(*n* = 852)	0.02	(*n* = 851)	0.61	(*n* = 853)	0.14	(*n* = 848)	0.31
Left-wing	90.4		81.4		71.1		91.8		97.0		95.8		82.6		85.2		64.8	
Center	83.5		70.7		71.4		88.4		96.0		91.9		80.1		80.4		62.1	
Right-wing	76.6		70.6		61.5		85.1		97.2		88.6		78.9		78.1		56.4	
Social class	(*n* = 917)	0.16	(*n* = 912)	0.41	(*n* = 912)	0.59	(*n* = 919)	0.12	(*n* = 922)	0.48	(*n* = 919)	0.37	(*n* = 917)	0.56	(*n* = 919)	0.40	(*n* = 914)	0.19
High	89.7		78.9		73.4		92.3		96.9		95.0		81.5		85.0		67.9	
Middle	86.5		78.9		69.3		90.5		95.9		93.8		78.0		82.6		60.1	
Low	84.4		75.1		71.5		87.5		97.6		92.3		81.1		80.9		63.9	

* *p* values were obtained from chi-squared (χ^2^) test. ** TL = Traffic light.

**Table 3 ijerph-19-08561-t003:** Prevalence ratios (95% CI) of support for food policies, yielded with Poisson regression models.

	Advertising Policies		Labeling Policies		Composition Policies		Provision and Sale Policies	
Sociodemographic Characteristics	Adjusted Model *	*p*	Adjusted Model *	*p*	Adjusted Model *	*p*	Adjusted Model *	*p*
Sex		0.00		0.03		0.03		0.09
Men	1		1		1		1	
Women	1.21 (1.06–1.37)		1.07 (1.01–1.13)		1.18 (1.02–1.36)		1.13 (0.98–1.30)	
Age (years)		0.02		0.50		0.34		0.01
18–29	1		1		1		1	
30–44	1.01 (0.78–1.31)		0.94 (0.85–1.04)		0.90 (0.69–1.17)		1.16 (0.82–1.64)	
45–64	1.28 (1.00–1.63)		1.00 (0.91–1.10)		0.96 (0.74–1.24)		1.53 (1.11–2.13)	
>65	1.27 (0.93–1.72)		0.98 (0.85–1.13)		1.15 (0.82–1.61)		1.52 (1.03–2.24)	
Educational level		0.36		0.17		0.49		0.10
Primary	1		1		1		1	
Secondary	1.10 (0.89–1.36)		0.99 (0.92–1.08)		0.93 (0.75–1.16)		1.07 (0.85–1.34)	
University	1.00 (0.78–1.29)		0.92 (0.82–1.02)		0.85 (0.64–1.12)		0.87 (0.66–1.16)	
Occupational status		0.88		0.89		0.72		0.19
Gainfully employed	1		1		1		1	
Pensioner	0.98 (0.79–1.20)		1.01 (0.91–1.13)		1.06 (0.82–1.36)		1.17 (0.93–1.46)	
Unemployed/unremunerated work	1.06 (0.89–1.27)		1.02 (0.94–1.10)		0.95 (0.75–1.19)		0.91 (0.72–1.16)	
Student	0.99 (0.61–1.62)		0.94 (0.77–1.15)		0.77 (0.44–1.36)		0.57 (0.24–1.36)	
Ideology		0.20		0.65		0.03		0.16
Left-wing	1		1		1		1	
Center	0.93 (0.81–1.05)		0.97 (0.92–1.04)		0.88 (0.76-1.02)		0.94 (0.81–1.09)	
Right-wing	0.83 (0.66–1.04)		0.97 (0.88–1.07)		0.72 (0.56–0.94)		0.84 (0.69–1.02)	
Social class		0.19		0.13		0.27		0.01
High	1		1		1		1	
Middle	0.89 (0.75–1.05)		0.92 (0.84–1.00)		0.85 (0.70–1.04)		0.74 (0.60–0.90)	
Low	0.85 (0.72–1.02)		0.97 (0.90–1.06)		0.87 (0.71–1.07)		0.84 (0.69–1.02)	

* Adjusted for all the variables included in the model.

## Data Availability

The datasets used during the current study are available from the corresponding author on reasonable request at mroyo@isciii.es.

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
