# Peer review of "Public Opinion on Food Policies to Combat Obesity in Spain"

_ijerph, 2022, doi:10.3390/ijerph19148561_

Round 1
Reviewer 1 Report
This research focus on the public opinion on food policies to combat obesity in Spain, and draw a conclusion that the authorities enjoy the support of the Spanish public as regarding implementing food policies proposed by experts and overcoming the resistance of sectors opposed to such measures. I have several concerns about this manuscript.
1. More details about the factors associated the public opinion on food policies are needed in the Introduction.
2. The authors found several factors influencing the public opinion, such as sex, age, ideology. However, that how these associations promote an improvement of food polices is still unknown, which is the core of your research.
Author Response
Dear editors,
We appreciate and are grateful for the comments and suggestions of the reviewers, which have served to improve the manuscript. Below there is a point-by-point description of the changes made in the manuscript in response to the reviewers comments. We’ve introduced the amendments in track changes in the manuscript.
Comments and Suggestions for Authors
Reviewer 1
This research focus on the public opinion on food policies to combat obesity in Spain, and draw a conclusion that the authorities enjoy the support of the Spanish public as regarding implementing food policies proposed by experts and overcoming the resistance of sectors opposed to such measures. I have several concerns about this manuscript.
- More details about the factors associated the public opinion on food policies are needed in the Introduction.
As suggested, we have provided more details about the factors associated the public opinion on food policies (see page 2, third paragraph).
We really appreciate the contributions of the reviewers to improve the quality of our manuscript.
Sincerely yours,
Cristina Cavero, on behalf of all authors

Reviewer 2 Report
Thank you for the opportunity to review the revised version of this manuscript. The authors have improved its content. My final comment is that Tables 2 and 3 should be one table as Table 3 does not provide any new stand-alone content. It should also indicate where the p-values have been derived from.
Author Response
Dear editors,
We appreciate and are grateful for the comments and suggestions of the reviewers, which have served to improve the manuscript. Below there is a point-by-point description of the changes made in the manuscript in response to the reviewers comments. We’ve introduced the amendments in track changes in the manuscript.
According to your instructions, we would like to mention that we have revised all the tables and made them into a table format. Additionally, we have changed Materials and Methods section subtitles 2.1 and 2.2 as specified.
Comments and Suggestions for Authors
Reviewer 2
Thank you for the opportunity to review the revised version of this manuscript. The authors have improved its content. My final comment is that Tables 2 and 3 should be one table as Table 3 does not provide any new stand-alone content. It should also indicate where the p-values have been derived from.
Thank you for your comments. As suggested, table 2 and 3 are now in the same table (see Table 2). In order to make results more concise and less repetitive, we had already grouped some policies in tables 2 and 3, except for the head of advertising policies, where each proposed measure focused on a different and relevant aspect of the policies with different levels of public support (see changes throughout the manuscript). As suggested, we have added how p-values have been calculated (see at the bottom of the table). Table 3 is what previously was table 4.
We really appreciate the contributions of the reviewers to improve the quality of our manuscript.
Sincerely yours,
Cristina Cavero, on behalf of all authors

Round 2
Reviewer 1 Report
I thought the authors misunderstood the comment 2 in review report. Although the authors identified several factors affecting the public opinion, how these associations promote an improvement of food polices is still elusive. Is there any similar research focusing on the same question? Which suggestions did they provide to enhance the recognition of food polices? According to your study, how can you promote the public to accept food polices? Your own thoughts are needed to add to the discussion, which improves the significance of your study.
Author Response
As suggested, we have made suggestions to promote the public to accept food policies, taking into account the suggestions made in similar research.
See changes in the end of the discussion section (page 12, third paragraph) as follows:
Understanding differences in the support for each food policy across sociodemographic characteristics could help to design communication campaigns aimed to less supportive populations groups to counter the dominant narrative, focusing on the external causes of obesity (social and commercial factors) and solutions, as suggested by Mazzocchi et al. In addition, to promote the public to accept food policies, we suggest developing education programs on nutrition for the less educated and supportive groups. Furthermore, to foster food policies and overcome resistance it will be neccessary to create aliances between public health professionals and civil society organizations. Enacting food policies will, in turn, change public attitudes as a result of the preference for the current state of affairs, the so called status quo bias, or by a process of cognitive dissonance whereby attitudes follow behaviour.
We really appreciate the contributions of the reviewers to improve the quality of our manuscript.
Sincerely yours,
Cristina Cavero, on behalf of all authors

This manuscript is a resubmission of an earlier submission. The following is a list of the peer review reports and author responses from that submission.
Round 1
Reviewer 1 Report
Thank you for the opportunity to review this manuscript. The study examines the extent to which a sample of Spanish adults support food policies. Whilst the survey design and analytical approach appear robust, there are several areas where this manuscript could be improved. These include:
1. The English expression could be improved throughout the manuscript. For example, the word 'enjoy' is not well placed in the phrase 'policies that enjoyed the highest level of support.'
2. The sampling approach is difficult to follow as it has been presented. For example, it is unclear how phone numbers can be randomly generated across different strata.
3. The results tables and text are repetitive and could be presented more concisely and effectively.
4. The discussion makes some good comparisons to the existing literature. Nonetheless, this is a cross-sectional study and many of the inferences being made in the discussion suggest causality. Moreover, the relevance of understanding differences in the support for food policies across sociodemographic characteristics remains unclear and could have been elaborated on.
Reviewer 2 Report
This article focus on the public opinions on food policies to combat obesity in Spain and draw a conclusion that the authorities enjoy the support of the Spanish public as regards implementing food policies proposed by experts and overcoming the resistance of sectors opposed to such measures. However, this manuscript is more likely to be a opinion poll, not a research article. Moreover, the authors didn't provide any suggestions about the food policies to combat obesity. Notably, the questionary used in the article is too simple. Therefore, more researches are needed to support your conclusion.